# Interactive Visualization and Simplified Pattern Discovery in the COVID-19 Open Research Dataset(CORD-19)

## Abstract

This work explores the use of Natural Language Processing based algorithms for Large Text Mining and Interactive Visualization for the COVID-19 Open Research Dataset (CORD-19) Dataset. We developed a series of easy to use online interactive text visualization based on different percentages of mined text data of diseases and chemical entities from the CORD-19 Dataset.This is to enable the study of patterns based on the frequency of entities in a very large dataset of about 2.6 million disease and chemical entities extracted from 31,376 papers. This will be useful to medical professionals, especially those who are not familiar with data mining techniques to interact with diseases, symptoms, drugs and chemicals texts entities to study patterns, relationships and trends to derive insights about the COVID-19 disease from publications about the disease and similar diseases. These extracted entities will also be made publicly available so that more work can be done with the dataset.

## 1 Introduction

In Wuhan, China, a novel and alarmingly contagious primary atypical (viral) pneumonia broke out in December 2019. It has since been identified as a zoonotic coronavirus, similar to SARS coronavirus and MERS coronavirus and named COVID-19(Liu et al., 2020). The World Health Organization (WHO) on March 11, 2020, has declared the novel coronavirus (COVID-19) outbreak a global pandemic(Cucinotta and Vanelli, 2020). Different measures are being taken globally to tackle the pandemic, one of them is the release of the CORD-19 Dataset.

On March 16, 2020, the Allen Institute for AI (AI2), in collaboration with partners at The White House Office of Science and Technology Policy (OSTP), the National Library of Medicine (NLM), the Chan Zuckerburg Initiative (CZI), Microsoft Research, and Kaggle, coordinated by Georgetown University's Center for Security and Emerging Technology (CSET), released the first version of CORD-19.(Wang et al., 2020)

The COVID-19 Open Research Dataset (CORD-19) is a growing resource of scientific papers on COVID-19 and related historical coronavirus research. CORD-19 is designed to facilitate the development of text mining and information retrieval systems over its rich collection of metadata and structured full text papers(Wang et al., 2020)

This dataset is intended to mobilize researchers to apply recent advances in natural language processing to generate new insights in support of the fight against this infectious disease. The corpus is updated regularly as new research is published in peer-reviewed publications and archival services like bioRxiv, medRxiv, and others

This work aims to provide a simple interface for medical professionals via an interactive web-based visualization tool using Scattertext, a Python text visualization library. This will provide a platform to study patterns, relationship based on frequencies of disease and chemical named entities extracted using scispaCy, a python Natural Language Processing Library.

scispaCy is a specialized NLP library for processing biomedical texts which builds on the robust spaCy library scispaCy models are useful on a wide variety of types of text with a biomedical focus, such as clinical notes, academic papers, clinical trials reports and medical records(Neumann et al., 2019)

In order to analyse the result of disease and chemical entities extraction of about 2.6 million tokens/phrases, we explored the use of text data visualization.

Finding words and phrases that discriminate categories of text is a common application of statistical Natural Language Processing(NLP). A wide range

of visualizations have been used to highlight discriminating words– simple ranked lists of words, word clouds, word bubbles, and word-based scatter plots These techniques have a number of limitations. For example, the difficulty in comparing the relative frequencies of two terms in a word cloud, or in legibly displaying term labels in scatterplots.(Kessler, 2017)

Scattertext is an interactive, scalable tool which overcomes many of these limitations. It is built around a scatterplot which displays a high number of words and phrases used in a corpus. Points representing terms are positioned to allow a high number of unobstructed labels and to indicate category association. The coordinates of a point indicate how frequently the word is used in each category.(Kessler, 2017)

## 2 Method

For this analysis, The custom licence subset of the CORD-19 Data set was downloaded on the 23rd of April,2020. This PDF subset of the data containing 31376 publications was used.

During the data preprocessing,stop words and single lettered words were retained because of chemical entities with single lettered symbols E.g. K for Potassium

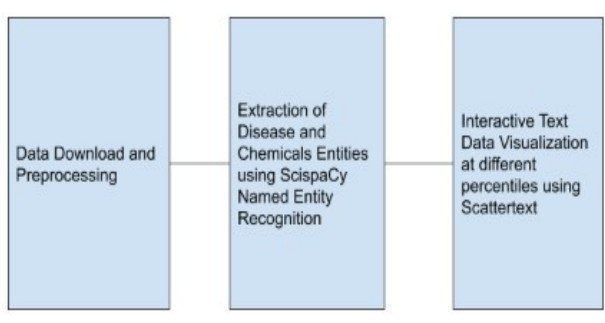

Figure 1: Cord-19 Data Mining Process

Scispacy Named Entity Recognition Model (en_ner_bc5cdr_md) trained on the BioCreative V Chemical-Disease Relations (BC5CDR) corpus which consists of 1500 PubMed articles with 4409 annotated chemicals, 5818 diseases and 3116 chemical-disease interactions(Li et al., 2016) was used to extract entities related to diseases and chemicals in the CORD 19 dataset.

The computation for the named entity extraction process was run in google colaboratory notebook GPU and the process took 8 hours to extract about 2.6 Million disease and chemical entities

Extracted entities of disease and chemicals were

then loaded as corpus for visualization with scattertext, only tokens with at least 100 tokens were .
Since the dataset is large, to aid exploration, the corpus was divided into 1,10,20,30,40,50,60,70,80,90 and 100 percentages. Insights that can be gleaned at each percentage of corpus differs based on word frequency

## 3 Results

Table1 shows the breakdown of entities per entity type

| Entity Name | Number of Extracted Tokens |
|---|---|
| Disease | 1371743 |
| Chemical | 1278831 |
| Total | 2650680 |

Table 1: Number of Disease and Chemical Entities mined from 31376 CORD-19 Publications

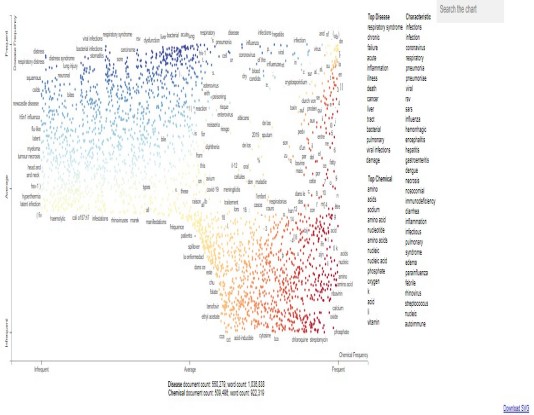

Figure 2: View of Html Interface for Interactive text visualization of disease and chemical entities

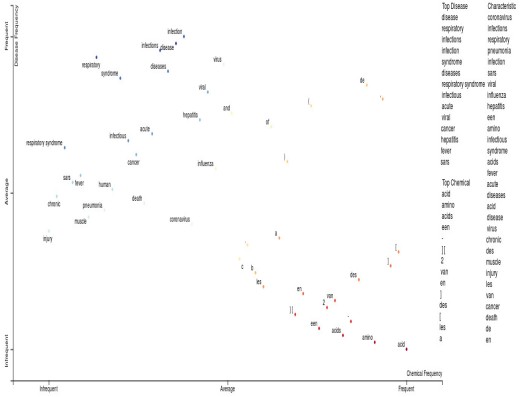

Figure 3: Interactive view of 1% of disease and chemicals entities corpus

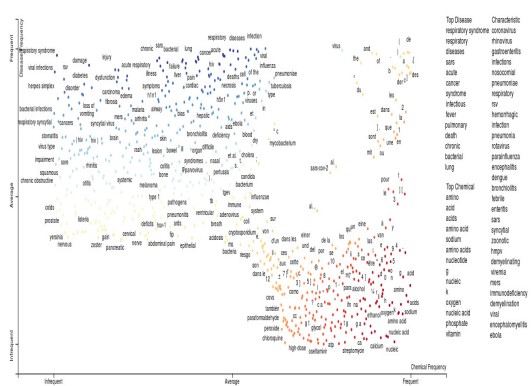

Figure 4: Interactive view of 10% of disease and chemicals entities corpus

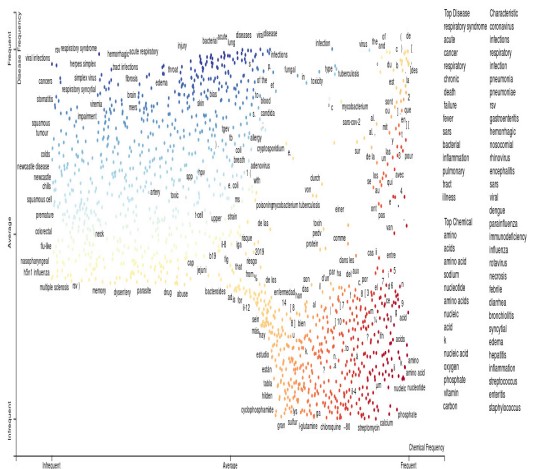

Figure 5: Interactive view of 20% of disease and chemicals entities corpus

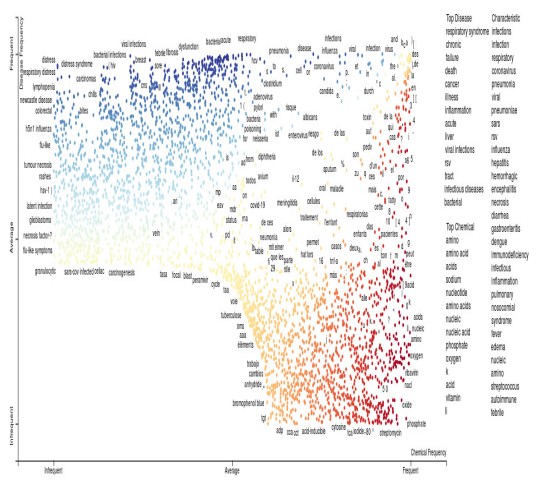

Figure 6: Interactive view of 50% of disease and chemicals entities corpus

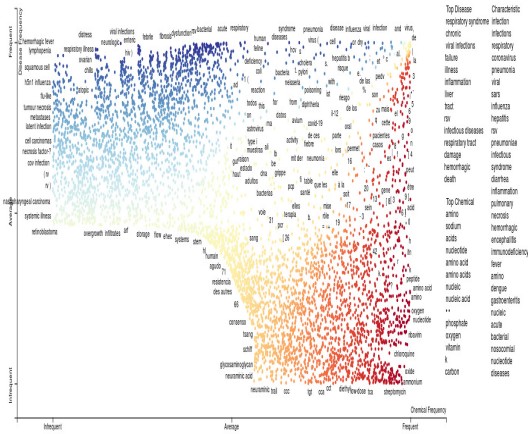

Figure 7: Interactive view of 80% of disease and chemicals entities corpus

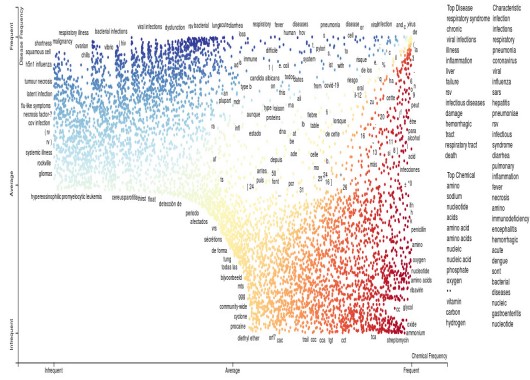

Figure 8: Interactive view of 100% of disease and chemicals entities corpus

## 4 Discussion

Each of the data point in the text visualization, when clicked shows the text data in different contexts from the CORD-19 dataset.There is also a search box where users of this tool can type in words related to diseases and chemicals they suspect may be significant to their study.E.g name of a particular medication or symptom.

Data points with deeper color tone implies that the particular word has many occurrences in the mined dataset.

Links to the deployed interactive web visualization will be included as embedding in the final submission(This is to preserve anonymity)

## 5 Conclusion

This work presented a text visualization method to interact with extracted diseases and chemical entities data that was extracted using Named Entity Recognition from the COVID-19 Open Research

Dataset.The interactive web interface is intuitive and can be used by anyone who understand diseases and chemicals to explore the data.This can be used by medical professionals to have mastery of the dynamic pattern in the COVID-19 management by understanding and exploring patterns and relationships in an effortless manner.

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
