# OpenReview forum: "Interactive Visualization and Simplified Pattern Discovery in the  COVID-19 Open Research Dataset(CORD-19)"
_EMNLP/2020/Workshop/NLP-COVID — Submitted to NLP-COVID19-EMNLP_

### Official Review · AnonReviewer2 · 2020-09-07
**Work-in-progress, not ready for publication**

**Rating:** 3
**Confidence:** 4

**Review:**

I found this article reports work-in-progress and I think it is still not ready for publication. Even though a visualization tool could certainly help medical professionals, no URL is provided to test the system (only screenshots). The author does not discuss its use case (beyond mentioning "the study of patterns") nor provides the perspective from a medical professional: How could a medical doctor reuse these "patterns" provided with the Scattertext library? How a healthcare professional would know which "patterns" to explore and look for with more detail in the literature?
Moreover, there are several typographic mistakes and grammar infelicities that seem like something authors dashed off. Some sentences or paragraphs could be rewritten and condensed for sake of clarity.

I provide some comments below.

##### 1. Abstract:
-L. 20, 26... -> What do authors mean by "patterns"? Co-occurrence of mentions of diseases and drug entities (which might reveal a therapeutic relation)?

##### 2. Introduction:
-L. 38: "... named COVID19" -> This is not correct; please, change to "SARS-CoV-2". The coronavirus was named "SARS-CoV-2", whereas the disease was named "COVID-19". Idem in l. 40.

-The paragraphs starting in line 42 through l. 64 seem to revolve around the same idea; it seems more adequate to rewrite them into one more synthetic paragraph.

-L. 81: The reference to the Scattertext library (Kessler 2017), and the URL link to it, should be provided the first time it is mentioned, not in line 107.

L. 85: The reference to sciSpacy should be given when this library is mentioned for the first time. Moreover, the updated bibliographic reference is:

    ScispaCy: Fast and Robust Models for Biomedical Natural Language Processing
    Mark Neumann, Daniel King, Iz Beltagy, Waleed Ammar
    Proc. of the 18th BioNLP Workshop and Shared Task, 2019, August 1st, Florence, Italy, pp. 319–327

##### 3. Results:
-Figures 2-8 are too small to read the entities shown. I recommend the author to place them in an Appendix, in a larger size and with higher resolution.

##### 4. Discussion:
The author mention an "interactive web visualization", but no URL is provided to test it.

##### 5. Others (grammar, style...):

-Write a white space between "Dataset" and the opening bracket in the title.
There are similar problems throughout all the article: a white space is missing between the last character of some words and the following opening bracket; e.g. l. 40, "pandemic(Cucinotta and Vanelli, 2020)"
I think this is an issue related to the Latex code.

-"Dataset" is repeated in "COVID-19 Open Research Dataset (CORD-19) Dataset"

-Abstract: "easy to use" -> "easy-to-use"

-I think the CORD-19 dataset should be cited in the first mention (l. 44)

-Several sentences lack a dot at the end of the sentence: e.g. l. 64, l. 78, l. 90, l. 103...

-L. 118: "For this analysis, The custom license" -> the (lowercase)

-L. 137: "Scispacy Named Entity Recognition Model" -> "The sciSpacy Named Entity Recognition Model"

-In general: the author is encouraged to split long sentences into short text fragments. For instance, the paragraph starting at l. 137 is made up of only 1 sentence. This could be fragmented in shorter sentences for the sake of clarity.

-L. 146: google -> Google

-L. 151: "...100 tokens were ." -> something is missing.

---

### Official Review · AnonReviewer1 · 2020-09-20
**An exploration on CORD-19 lacking thorough evaluations**

**Rating:** 5
**Confidence:** 4

**Review:**

This study provides an online visualization tool for discovering CORD-19 dataset. It applies scispaCy to annotate the concepts mentioned in the papers and then applies scattertext for visualization.

My major concern is none of these steps was evaluated. The pre-trained models in scispacy were trained using the biomedical datasets, but may not capture COVID-19 specific entities effectively. Are there any evaluations on the precision and recall of the models specifically for COVID-19 related entities? Similarly, the identified relations were not evaluated either.

In addition, the paper does not provide the URL of the tool for evaluation. Given the significant amount of NER time, how frequent is the tool updated? Are APIs available?

Also, the provided functions are limited. For instance, CORD-19 has papers that are not on COVID-19 (e.g., MERS). Visualizing the entities and relations on the whole dataset may not accurately represent the characteristics for COVID-19. Providing more filtering functions would be helpful.

The representation is also confusing. For instance, why providing 6 separate figures just showing different percentages of entities? A figure with sub-figures on selected percentages would be sufficient and the remaining figures could be used for demonstrating other functions.

---

### Official Review · AnonReviewer3 · 2020-09-21
**An interesting idea, but a more detailed explanation is required**

**Rating:** 4
**Confidence:** 4

**Review:**

This paper describes the development of a visualization tool that provides an overview of domain-specific named entities extracted from a subset of the COVID-19 Open Research Dataset (CORD-19). A pre-trained NER model from the ScispaCy library was used to extract named entities of CORD-19 articles, which are then visualized by the tool.

In general, the idea of providing a visualization tool for efficiently navigating CORD-19 is interesting. However, the author did not provide a way of accessing the tool (e.g. via weblink), making an extensive evaluation more difficult. Furthermore, the description of the proposed tool is vague.

Overall, I think the paper needs to be clearer in describing the tool and its functions, and an online access to the tool would be very helpful for a proper review.

Comments:
1. In many occurrences, text is not whitespace-separated from the reference (e.g. line 38, 41, 91).
2. I found multiple sentences without proper punctuation, which makes reading the paper more difficult (e.g. line 92, 78, 88, 160).
3. In line 120, the term “dataset” is spelled as “data set”.
4. In line 151, the sentence is incomplete.
5. None of the figures (1-8) shown in the paper are mentioned in the text, and a short explanation would be helpful. Furthermore, figures 2-8 are very hard to read in the current resolution. Also, what do the colors in the plots (blue, yellow, red) refer to?
6. A citation at the end of a sentence should be placed before punctuation (e.g. line 115, 107, 64).